# Photophoretic spectroscopy in atmospheric chemistry – high sensitivity measurements of light absorption by a single particle

Nir Bluvshtein, Ulrich K. Krieger, Thomas Peter

Institute for Atmospheric and Climate Science, ETH Zurich, 8092, Switzerland

*Correspondence to*: Nir Bluvshtein (nir.bluvshtein@env.ethz.ch)

**Abstract.** Light absorbing organic atmospheric particles, termed brown carbon, undergo chemical and photochemical aging processes during their lifetime in the atmosphere. The role these particles play in the global radiative balance and in the climate system is still uncertain. To better quantify their radiative forcing due to aerosol-radiation interactions, we need to improve process level understanding of aging processes, which lead to either 'browning' or 'bleaching' of organic aerosols. Currently available laboratory techniques aim to simulate atmospheric aerosol aging and measure the evolving light absorption, but suffer from low sensitivity and precision. This study describes the use of electrodynamic balance photophoretic spectroscopy (EDB-PPS) for high sensitivity and high precision measurements of light absorption by a single particle. We demonstrate the retrieval of time-evolving imaginary part of the refractive index for a single levitated particle in the range of $10^{-4}$ to $10^{-5}$ with uncertainties of less than 25% and 60%, respectively. The experimental system is housed within an environmental chamber, in which aging processes can be simulated in realistic atmospheric conditions and lifetime of days to weeks. This high level of sensitivity enables future studies to explore the major processes responsible for formation and degradation of brown carbon aerosols.

## 1 Introduction

Most radiative transfer schemes in climate models treat organic aerosol, a major subset of atmospheric aerosols that comprise 20-90% of the total particulate mass (Kanakidou et al., 2005; Zhang et al., 2007), as non-absorbing in the UV-Vis wavelength range, attributing them with a negative (cooling) radiative effect. However, light absorbing organic aerosol, termed brown carbon (BrC), with wavelength dependent light absorption ($\lambda^{-2} - \lambda^{-6}$) in the UV-Vis wavelength range (Chen and Bond, 2010; Hoffer et al., 2004; Kaskaoutis et al., 2007; Kirchstetter et al., 2004; Lack et al., 2012b; Moosmuller et al., 2011; Sun et al., 2007), may be the dominant light absorber downwind of urban and industrialized areas and in biomass burning plumes (Feng et al., 2013). Recently, it has been shown that including BrC absorption properties in radiative transfer models lead to a stronger wavelength dependency of light absorption by aerosols and to significant changes in the overall effective radiative forcing from aerosol-radiation interactions (ERFari) (Feng et al., 2013; Lack and Cappa, 2010). Atmospheric aging processes of organic aerosol can lead, through complex mechanisms, to formation of light absorbing compounds ('browning') or to their degradation ('bleaching'). The accurate characterization of these processes is one of the main open questions in atmospheric

chemistry research and has been the focus of many recent studies (see *Laskin et al.* (2015) for a review). Although significant advances have been made, the contribution of BrC to anthropogenic radiative forcing still poses significant uncertainty. The estimated ERFari attributed to BrC is 0.1 to 0.25 W m$^{-2}$, offsetting 10 to 25% of the global mean aerosol cooling effect ($-1.1^{-0.1}_{-1.95}$ W m$^{-2}$) (Bond et al., 2013; Brown et al., 2018; Feng et al., 2013; Myhre et al., 2013). On a regional scale surrounding mega-cities and industrial areas, this value may be up to an order of magnitude higher (Feng et al., 2013), which nearly doubles the local warming effect caused by the increase in the $CO_2$ concentration. For this reason, it is imperative to (i) better understand the formation and degradation of BrC aerosols resulting from chemical and photochemical aging processes (Dasari et al., 2019; Drozd and McNeill, 2014; Hems and Abbatt, 2018; Lambe et al., 2013; Lee et al., 2013, 2014; Marrero-Ortiz et al., 2019; Powelson et al., 2014; Romonosky et al., 2015; Saleh et al., 2014; Schnitzler and Abbatt, 2018; Zhao et al., 2015; Zheng et al., 2013), (ii) quantify the BrC wavelength dependent light absorption and (iii) identify the main molecular species responsible for this absorption.

Laboratory studies simulating BrC formation and degradation mechanisms generally take one of two approaches to quantify wavelength dependent light absorption in the UV-Vis wavelength range, which is described using the imaginary part ($\kappa$) of the complex refractive index (CRI; $m = n + i\kappa$). The first approach uses bulk liquid phase or gas–liquid multiphase experiments to simulate atmospheric chemical processes. This is then followed by UV-Vis spectroscopy absorption measurements (Nguyen et al., 2012; Nozière et al., 2010; Nozière and Córdova, 2008; Updyke et al., 2012). The advantages are high sample volume available for analysis, and the extremely high sensitivity of UV-Vis absorption spectroscopy. The disadvantage is that supersaturated conditions often encountered in atmospheric aerosol particles are impossible to generate in a bulk volume. This is an important disadvantage of the bulk approach as chemical activity, viscosity and diffusivity under supersaturated conditions may alter the chemical aging significantly.

The second approach uses an environmental chamber, reactor, or flow tube to reproduce atmospheric processes. In this context, particles are generated by aerosolization into the experimental volume or by gas phase chemistry leading to reduced volatility of precursor compounds with subsequent gas-to-particle conversion. These aerosols are then subjected to chemical or physical aging processes using radiation, relative humidity and reactive gaseous components. The aged material may then be collected on filter substrates, extracted with a liquid solvent and analyzed by UV-Vis spectroscopy. Alternatively, aerosols may be measured with a suite of particle-ensemble, flow-through optical methods, such as cavity enhanced and photoacoustic spectroscopy, for both direct and indirect measurements of light absorption (Flores et al., 2014a, 2014b; He et al., 2018; Nakayama et al., 2013). To maintain particle size and concentration high enough for analysis using these techniques and to simulate atmospheric exposure of up to several days, precursor concentrations often far exceed those of the ambient atmosphere. This difference in environment between the reactor and the real atmosphere may affect the final distribution of products after multi-generation chemical aging, which may lead to misleading interpretations of the experimental results. An advantage of the optical instruments compared with the filter extraction techniques is the independence from solubility, extraction efficiency and solvent matrix effects. Additionally, particles are measured *in-situ* and with higher time resolution. Disadvantages are the relatively low sensitivity to absorption and, for some instruments, the frequent need for calibration (Lack

et al., 2012a). As a result, $\kappa$ for BrC aerosols is often reported with values near the limit of quantification of the retrieval technique, or that actually conflicts with results from analyses of similar chemical systems (Liu et al., 2013, 2012; Nakayama et al., 2013; Ofner et al., 2011) and with uncertainties that can exceed 100% (Bluvshtein et al., 2016; Flores et al., 2014a, 2014b; He et al., 2018; Lack et al., 2012b; Lavi et al., 2013; Nakayama et al., 2010, 2013; Trainic et al., 2011; Washenfelder et al., 2013).

To make significant progress, reduce uncertainties, and resolve contradictions, this study aims to extend the approach of electrodynamic balance photophoretic spectroscopy (EDB-PPS) for high sensitivity and high precision measurements of UV-vis light absorption by a single particle, in an environmental chamber, exposed to realistic atmospheric aging processes.

Photophoresis describes the optical forces acting on an illuminated particle. Direct photophoresis (or "radiation pressure" force; $F_{rp}$), always in the direction of light propagation (away from the light source) and indirect photophoresis ($F_{ph}$), a less recognized force, central to this work, that strongly depends on the absorption of light. Unlike radiation pressure, indirect photophoresis is a result of an uneven temperature distribution on the surface of the particle resulting from absorption. It acts through momentum transfer with the surrounding gas molecules, and thereby it is temperature and pressure dependent. To resolve the direction (away from or towards the light source) and the magnitude of the indirect photophoresis force one needs to determine the internal electric field distribution (Bohren, 1983; Mackowski, 1989). Pope et al. (Pope et al., 1979) and Arnold et al. (Arnold et al., 1980) conceived the idea of PPS on a single levitated particle and described the spectrally dependent ratio of the measured photophoretic force to the gravitational force on the particle. Building upon their seminal work with following developments in the mathematical description of indirect photophoresis (Beresnev et al., 1992; Mackowski, 1989; Rohatschek, 1995), computational advances in Mie theory and internal field calculation (Hovenier, 2000), we extend the use of PPS. Here we describe how PPS can be used for high sensitivity retrieval of the imaginary part of the refractive index of organic aerosol proxy particles levitated in an environmental chamber EDB and subject to aging processes. The EDB is a well-established tool in atmospheric science, used for the study of thermodynamic and chemical properties of single levitated particles (Krieger et al., 2000; Steimer et al., 2015; Tang and Munkelwitz, 1994; Zardini et al., 2008). Its high sensitivity to changes in the net vertical force acting on the particle makes it ideal for measurements of photophoresis, i.e. the miniscule optical forces acting on the levitated particle as a result of its interaction with light.

## 2 Experimental

### 2.1 Electrodynamic balance

Over 100 years ago Robert Millikan and Harvey Fletcher, in their famous oil drops experiment, showed that tuning the electric potential between two capacitor plates, required to levitate a charged particle, can be used as a balance with a sensitivity of about $10^{-13}$ g. Today, the electrodynamic balance (EDB) is an established tool used to derive thermodynamic and physical information of a single, levitated particle. The EDB used in this study (Figure 1) is based on the double-ring design characterized by Davis et al. (Davis et al., 1990) and described in previous publications (Colberg et al., 2004; Steimer et al.,

2015). Only a brief description follows. The EDB is hosted within a 180 cm$^3$ environmental chamber, which allows temperature and pressure regulation with precision of ±0.02 K and ±2 hPa. Mass flow controllers are used to regulate gas flow rate and composition to 5 ccm of dry $N_2$. A single-droplet generator (Hewlett-Packard 51633A ink jet cartridge) is used to inject an electrically charged diluted aqueous solution of the sample material to the center of the trap. After complete loss of water, the resulting particle (typically 7-13 µm in radius) is balanced by a DC voltage that is regulated with a 25 Hz automated video feedback loop. The DC voltage ($U$) that is applied between the EDB endcap electrodes, to hold the particle in the center of the trap, is proportional to the net vertical force acting on the particle, with sensitivity as low as 0.1%, which is equivalent to a mass of about $10^{-15}$ g. This is commonly used to measure small changes in particle mass due to loss or uptake of gaseous components by the particle (Steimer et al., 2015; Tang and Munkelwitz, 1994; Zardini et al., 2008). In this study, we alternately illuminate a suspended particle with a 473 nm, 33 mW mm$^{-2}$ (±15%) diode laser (Laser Quantum, gem 473) at 25% duty 40-s cycles (Figure 1b). Variations in the DC voltage from 'laser off' to 'laser on' ($\Delta U/U_0$) in each cycle were used to measure changes in the net vertical force induced by illumination of the particle:

$$\frac{\Delta U}{U_0} = \frac{U_{\mathrm{on}} - U_{\mathrm{off}}}{U_{\mathrm{off}}} = \frac{F_{\mathrm{ph}} + F_{\mathrm{rp}}}{F_g + F_s} . \quad (1)$$

Here, $F_{\mathrm{ph}}$ and $F_{\mathrm{rp}}$ are the photophoretic and radiation pressure forces. The subscripts "on" and "off" relate to the light source. Gravity ($F_g$) and Stokes drag ($F_s$) caused by the gas flow, are constant at the time scale of the measurement, and are given by

$$F_g = \frac{4}{3}\pi a^3 \rho_p g , \quad (2)$$

$$F_s = \frac{6\pi a \eta q_v}{C_c S} , \quad (3)$$

where $a$ and $\rho_p$ are radius and mass density of the levitated particle, $g$ is the standard acceleration due to gravity, $\eta$ is the gas dynamic viscosity, $q_v$ is volumetric flow rate, $S$ is a characteristic flow cross section and $C_c$ is the Cunningham slip correction factor (Kim et al., 2005). This formulation enables the direct measurement of the sum of the optical forces ($F_{ph} + F_{rp}$), which are related to the particle size and CRI. After determining the particle size and the real part of the refractive index $n$ (Section 2.2.), the sum $F_{ph} + F_{rp}$ is iteratively calculated by varying $\kappa$ to minimize the difference between the measured and calculated $\Delta U/U_0$.

The following section describes how the particle size and real refractive index are determined from high-resolution light scattering measurements and Section 2.3 describes how $F_{ph}+F_{rp}$ is finally calculated and used to retrieve the imaginary part of the CRI.

## 2.2 Determination of size and real refractive index

Mie resonance spectroscopy is used to simultaneously retrieve the particle's radius ($a$) and real part ($n$) of the CRI defined as:

$$a = \frac{\lambda \chi}{2\pi} , \quad (4)$$

and

$$n(\lambda) = n_D + m_1 \times \left(\frac{1}{\lambda^2} - \frac{1}{\lambda_D^2}\right) . \quad (5)$$

Here, $n(\lambda)$ is the wavelength-dependent real refractive index, $n_D$ is the refractive index at the sodium D-Line, $m_1$ is a dispersion coefficient and $\chi$ is the size parameter. Polynomial regression parameters between the refractive index and size parameter of transverse electric (TE) and transverse magnetic (TM) mode resonances were calculated (Lam et al., 1992; Preston and Reid, 2013, 2015) and a look-up table of these parameters was generated for all possible TE and TM resonances with order numbers of 3 to 8 and mode numbers of 26 to 180. We obtained high-resolution spectra by illuminating the levitated particle with a tunable diode laser at both parallel and perpendicular linear polarizations (TDL, New Focus, model Velocity 6312) in the range $\lambda$ = 765–781 nm and recording the elastic light scattering at $\pi/2$ angle (Steimer et al., 2015) (Figure 1a,3). Then, $a$, $n_D$ and $m_1$ are retrieved by minimizing the difference between measured and calculated wavelengths of the Mie resonances over the three dimensional parameter space.

### 2.3 Photophoretic spectroscopy

Direct photophoresis or radiation pressure is readily calculated from

$$F_{\mathrm{rp}} = (Q_{\mathrm{ext}} - Q_{\mathrm{bs}}) \times \frac{\pi a^2 I}{c}, \quad (6)$$

where $I$ is the radiant flux density (in W/m$^2$) and $c$ is the speed of light. $Q_{\mathrm{ext}}$ and $Q_{\mathrm{bs}}$ are Mie extinction and back scattering efficiencies (unitless). We use the Mie Matlab functions developed by Mätzler (2002) to calculate the efficiencies. Indirect photophoresis ($F_{\mathrm{ph}}$) is directed away from the light source for a highly absorbing particle (positive photophoresis) but towards the light source for low absorptivity (negative photophoresis). This is a result of the structure of the internal electric field within a spherical particle interacting with radiation. Multiple refractions and internal reflections lead to size dependent, nano-focusing of the incident beam within the particle volume. For particles larger than the wavelength of the incident light, the energy is "focused" closer to the non-illuminated side of the particle. In highly absorbing particles, however, most of the energy is absorbed by the illuminated hemisphere of the particle, heating it more than the "dark" hemisphere. Therefore, a key parameter determining the direction and amplitude of $F_{\mathrm{ph}}$ is the temperature asymmetry parameter ($J$), resulting from the uneven internal electric field and, consequently, uneven temperature distribution (Yalamov et al., 1976). To calculate $J$, one can use an integration of the source function over the particle volume:

$$J(\chi, m) = 3n\kappa\chi \int_0^1 \int_{-1}^1 B(t,\mu) t^3 \mu d\mu\, dt\ . \quad (7)$$

Here $B(t,\mu)$ is the dimensionless electric field distribution inside the particle, $t$ (fraction of $a$) and $\mu = \cos\theta$ are the spherical coordinates. Simply put, $J$ indirectly describes the temperature gradient between the illuminated and the "dark" side of the particle surface.

Unlike radiation pressure, $F_{\mathrm{ph}}$ acts through the presence of gas molecules around the illuminated particle. Impaction and reflection of the surrounding gas molecules, and consequent momentum transfer with the particle's surface, is temperature dependent and thus leads to a net force directed from the warmer to the colder particle hemisphere. Indirect photophoresis is also strongly pressure ($p$) dependent. It reaches its maximum value at pressures, where the Knudsen number (Kn = $L/a$) is unity, i.e. where the gas mean free path ($L$) is comparable to the radius of the particle ($a$). In the free molecular regime, i.e. Kn

>> 1, $F_{ph}$ is proportional to $p$, whereas in the continuum regime, i.e. Kn << 1, $F_{ph}$ is inversely proportional to $p$. Rohatschek (1995) provided a pressure dependent model of $F_{ph}$ interpolating previous formulations at the two pressure regime limits. His approach provides a convenient estimate of $F_{ph}$ interpolating between the free molecular and continuum limits:

$$F_{ph} = \frac{2F_{max}}{p/p_{max} + p_{max}/p}, \quad (8)$$

with

$$F_{max} = D\frac{a^2 JI}{k_p}\sqrt{\frac{\alpha_T}{2}}, \quad (9)$$

$$p_{max} = D\frac{3T}{\pi a}\sqrt{\frac{2}{\alpha_T}}, \quad (10)$$

where $\alpha_T$ is the thermal accommodation coefficient, $k_p$ is the particle thermal conductivity and $T$ is the gas temperature away from the particle surface. Further, $D$ relates to gas phase parameters as follows:

$$D = \frac{\pi \hat{c}\eta}{2T}\sqrt{\frac{\pi C_s}{3}}, \quad (11)$$

where $C_s$ is the thermal slip coefficient and $\hat{c}$ is the mean thermal velocity of the gas molecules

$$\hat{c} = \sqrt{\frac{8RT}{\pi M}}, \quad (12)$$

in which $R$ is the gas constant and $M$ is the gas molar mass.

Our experimental set up (particle radius of 7-13 µm and pressure range of 400-800 mbar) is limited to Kn = 0.0075-0.03. We are therefore constrained to a transition flow regime referred to as the slip-flow regime (typically $10^{-2}$ < Kn < $10^{-1}$), where $F_{ph}$ deviates significantly from Rohatschek's interpolation. Mackowski (1989) presented an analytical solutions of the spherical geometry heat conduction equation in three dimensions for calculating $F_{ph}$ in the slip-flow regime by adding a tangential velocity slip boundary condition (also referred to as thermal stress slip flow) to the continuum regime solution of Yalamov et al. (1976):

$$F_{\mathrm{ph}} = -\frac{4\pi C_s \eta^2 I a J}{\rho_g k_p T} \times \left[(1 + 3C_m \mathrm{Kn})\left(1 + 2C_t \mathrm{Kn} + 2\frac{k_g}{k_p}\right)\right]^{-1}. \quad (13)$$

The momentum exchange coefficient is taken as $C_m = 1.175 \pm 0.175$ (Reed, 1977), while the thermal slip ($C_s$) (Ivchenko et al., 1993) and temperature jump ($C_t$) (Loyalka, 1968) coefficients

$$C_s = \frac{3}{2} \times \left(\frac{0.4375 + 0.2084\alpha_T}{0.856 + 0.1092\alpha_T}\right), \quad (14)$$

$$C_t = \frac{5}{18} \times \frac{2 - \alpha_T}{\alpha_T} \times (1 + 0.1621\alpha_T), \quad (15)$$

are functions of the thermal accommodation coefficient $\alpha_T$. Here we used a value of $\alpha_T = 0.85 \pm 0.15$ to accommodate a range of values published for a variety of materials (Ganta et al., 2011; Li et al., 2001; Shaw and Lamb, 1999; Trott et al., 2007). It was shown that with increased absorption (i.e. steep interface temperature jump) or decreasing particle size, Equation (14) deviates from experimental measurements (Mackowski, 1989; Soong et al., 2010). A solution to this problem was

presented by Soong et al. (2010) who adopted a modified slip boundary condition from Lockerby et al. (2004). The authors developed the following correction to Mackowski's solution:

$$F_{\mathrm{ph_corr}} = F_{\mathrm{ph}} \times \left(1 + \frac{2C_m \mathrm{Kn}}{C_s}\right) . \quad (16)$$

From equation (16) it is clear that at Kn ≪ 1 the correction factor approaches unity. For the purpose of this study, i.e. low absorptivity particles and Kn = 0.0075-0.03, the correction is small, but we nevertheless apply it in our evaluation and term the corrected photophoretic force hereafter $F_{\mathrm{ph}}$.

As mentioned above, $J$ is the key parameter linking the particle's CRI to $F_{\mathrm{ph}}$. Mackowski (1989) also presented an expression for $J$ by analytical integration of equation (7):

$$J = -\frac{6n\kappa}{|m|^2\chi^3} \times \mathrm{Im} \sum_{N=1}^{\infty} \left( \frac{N(N+2)}{m} \left( \frac{c_{N+1} c_N^* R_N}{C_N} + d_{N+1} d_N^* R_{N+1} C_N^* \right) - \left( \frac{N(N+2)}{N+1} \left( \frac{c_{N+1} c_N^*}{C_N} + \frac{d_N d_{N+1}^*}{C_N^*} \right) + \frac{2N+1}{N(N+1)} d_N c_N^* \right) S_N \right) , \quad (17)$$

where the coefficients $C_N$, $R_N$ and $S_N$ are

$$C_N = \frac{N+1}{m\chi} - \frac{\psi_N'(m\chi)}{\psi_N(m\chi)} , \quad (18)$$

$$R_N = \frac{\mathrm{Im}(mC_N)}{\mathrm{Im}(m^2)} , \quad (19)$$

$$S_N = \frac{i}{2\mathrm{Im}(m^2)} \left\{ \chi(m + m^*|C_N|^2) - \left( m + 2(N+1)\frac{\mathrm{Re}(m^2)}{m} \right) R_N + (2N+1) m^* |C_N|^2 R_{N+1} \right\} , \quad (20)$$

$$c_N = \psi_N(m\chi)\tilde{c}_N , \quad (21)$$

$$d_N = \psi_N(m\chi)\tilde{d}_N . \quad (22)$$

Here, $\tilde{c}_N$ and $\tilde{d}_N$ are the Mie coefficients for the internal field, computed using the MatLab Mie routines by Mätzler (2002) and $\Psi_N$ is the Ricatti-Bessel function of order $N$. For clarity, the prime denotes the differentiation with respect to the argument in brackets and the superscript * denotes the complex conjugate. Figure 2 shows $F_{\mathrm{ph}}$ calculated with the above three models over a wide pressure range extending from the free molecular to the continuum flow regimes. From eq. (13) and Figure 2, one can show that Mackowski's formulation equals Rohatschek's solution at the continuum limit (i.e. Kn ≪ 1) only for particles that are good heat conductors compared with the surrounding gas (i.e. $k_g / k_p \ll 1$).

For applications involving slightly absorbing ($\kappa \leq 10^{-3}$) micron-sized, organic particles, the indirect photophoretic force is generally 1-2 orders of magnitude larger than radiation pressure but about 2-3 orders of magnitude lower than gravity. High sensitivity and stability of the EDB is therefore imperative for high sensitivity retrieval of $\kappa$ from EDB-PPS measurement.

## 3 Results and discussion

To test the methodology a slightly absorbing organic particle with known CRI and thermal properties is required. For this purpose, we selected PEG400 (polyethylene glycol with mean molecular weight of ≈ 400 g mol$^{-1}$) as a proxy. PEG400 has the advantage of being an organic, non-volatile liquid, miscible with water (needed for injection of a droplet into the EDB). Additionally, PEG400 has well characterized optical and thermodynamic properties (Francesconi et al., 2007; Han et al., 2008;

Marcos et al., 2018; Reyes et al., 2000), which we assume to be unchanged by addition of 0.23% wt (0.19% mole) of carminic acid (CA, Sigma-Aldrich). The imaginary part of the refractive index for this PEG400-CA solution ($\kappa = (1.394 \pm 0.05) \times 10^{-4}$) was determined with a simple Beer-Lambert setup composed of the 473-nm laser introduced in Section 2.1, a 1 mm cuvette, a power meter and using the following relations (Sun et al., 2007):

$$A = \log_{10}\frac{I_0}{I} = \alpha \times L \,, \tag{23}$$

$$\alpha = \frac{4\pi \times \kappa}{\lambda} \,, \tag{24}$$

where $A$ is the optical attenuation or absorption of a bulk sample with an optical path length $L$ and attenuation coefficient α at wavelength $\lambda$. Combining equations (23) and (24) leads to:

$$\kappa = \frac{A \times ln(10) \times \lambda}{4\pi \times L} \,. \tag{25}$$

High viscosity of the PEG400 leads to slight heterogeneity of the liquid in the cuvette. For this reason, we repeated this spectroscopic measurements with the laser beam (< 1 mm in diameter) crossing the cuvette at different positions on its surface. This led to the 3.6% uncertainty in the value of $\kappa$ stated above.

A droplet with 3% wt of the PEG400-CA solution in water was injected into the EDB under dry $N_2$ flow as described in Section 2.1. Following size stabilization and water evaporation, high-resolution Mie resonance spectra were measured to determine the particle real part of the CRI ($n_D$ = 1.4665, $m_1$ = 2745) and size ($a$ = 9.2906 µm). Figure 3 shows the measured TE and TM modes spectra and the fitted resonance peaks along with their identification by order ($l$) and mode ($n$) numbers. Figure 4 shows the response of the EDB to changes in the net vertical force due to illumination of the levitated particle ($\Delta U/U_0$) at different pressure values within the range of our experimental set up. Also shown in Figure 4 is the response calculated using the three models described in Section 2.3. It is clear that both the Mackowski and the Soong formulations, which are barely distinguishable within the resolution of Figure 4, fit the measured data very well, whereas the Rohatschek interpolation, which assumes ${}^{k_g}\!/_{k_p} \ll 1$ and does not account for slip-flow conditions, overestimates the response. The error bars on the measured data represent the standard deviation over five illumination cycles and the gray shaded area represents the uncertainty propagated through the Soong model calculation. The major contributor to the latter is a 15% uncertainty on the radiant flux, which is measured in our experiment with a power meter (nova-display, Ophir Optronics LTD) and a beam profiler (CMOS-1.001- Nano, CINOGY Technologies GmbH).

To further demonstrate the potential of the EDB-PPS approach in determining the imaginary RI with high sensitivity and precision, an additional particle from the same PEG400-CA batch, with radius of 12.858 µm was levitated and the response of the EDB to change in the net vertical force was recorded over about 16 hours of illumination cycles. To take advantage of the inverse pressure dependence of the photophoretic effect, this experiment was conducted at 400 hPa, which is at the lower limit of our experimental set up. Figure 5 shows the measured $\Delta U/U_0$ and the point-by-point retrieved $\kappa$, which decreases from about $1.25 \times 10^{-4}$ to about $0.01 \times 10^{-4}$ over the "laser on" time (i.e. 25% of the total experiment time). Both traces demonstrate a decay feature as a result of slow photolysis of the CA (Jørgensen and Skibsted, 1991). The initial retrieved value $\kappa$ =

$(1.251^{+0.315}_{-0.213}) \times 10^{-4}$ is about 20% lower than the value retrieved from the bulk cuvette experiment [$\kappa = (1.394 \pm 0.05) \times 10^{-4}$] but with overlapping uncertainty range. The overall uncertainty in retrieved $\kappa$ demonstrated in Figure 5b is estimated to be up to 25% for $\kappa \approx 10^{-4}$ and up to 60% for $\kappa \approx 10^{-5}$. The gray shaded area on the $\Delta U/U_0$ plot (Figure 5a) is the uncertainty in each $\Delta U/U_0$ data point determined from the standard deviation on the value of $U$ averaged during "laser off" and "laser on" stages of each individual illumination cycle. These experimental uncertainties, with values of $2\times10^{-5}$ - $10\times10^{-5}$ (unitless) together with point-by-point variability of up to $\pm10^{-4}$, are a result of the sum of system instabilities. These are independent of the value of $\Delta U/U_0$ and are a significant component limiting the sensitivity to $\kappa$ of this experimental approach. The experimental uncertainty contributes up to 10% uncertainty on the value of the retrieved $\kappa$ for values down to $\kappa \approx 2 \times 10^{-5}$ and about 30% for $\kappa < 2 \times 10^{-5}$. The uncertainty on the radiant flux used in the model calculation is the major limiting factor for the sensitivity of the experimental approach contributing about 15% to the uncertainty of the retrieved $\kappa$ down to $\kappa \approx 2 \times 10^{-5}$.

We determine the overall sensitivity of this approach to the imaginary RI (within the limitation of our experimental setup) to be in the range of $1 \times 10^{-5}$ - $2 \times 10^{-5}$. The accumulated uncertainty of $\kappa$ from all other parameters in Equations (14) and (17) is below 5% for $\kappa$ down to $\kappa \approx 2 \times 10^{-5}$. These include uncertainties in the particle size, real part of the CRI, density and thermal properties. This small contribution to the overall uncertainty demonstrates the usefulness of the EDB-PPS approach for the retrieval of the imaginary RI of organic particles with unknown composition following accurate Mie resonance spectroscopy (for determination of size and real RI) as long as assumptions of sphericity and homogeneity hold.

## 4 Conclusion

This study demonstrates the usefulness of the electrodynamic balance – photophoretic spectroscopy (EDB-PPS) technique to retrieve the imaginary component of the complex refractive index of a slightly absorbing organic particle levitated in an environmental chamber. We showed agreement between measurements and model calculation and reliable retrieval of the imaginary RI at a wavelength of 473-nm, in the range of $10^{-5} \leq \kappa \leq 10^{-4}$ with uncertainty of about 25% for $\kappa = 10^{-4}$and 60% for $\kappa = 10^{-5}$. The major limiting factor for sensitivity and precision is the uncertainty in the radiant flux within our setup. The range of environmental conditions allowed by the chamber are pressure of 400 – 800 hPa, RH of 0 – 90% and temperature of 200 – 300 K. This means that we can measure and understand heterogeneous chemistry and photochemical aging processes of a single particle in the full range of boundary layer conditions with atmospherically relevant gas concentration and residence time. The combination of high sensitivity and quantification level with the wide application range of the environmental chamber enables us to improve process level understanding of formation and degradation of BrC aerosols resulting from chemical and photochemical aging processes that is beyond the reach of previously available aerosol flow-through techniques. This study laid the needed foundations for future development of a new methodology aimed to simultaneously measure the evolution of light absorption and the molecular composition of proxies of atmospheric aerosols by coupling photophoretic spectroscopy to an electrodynamic balance – soft ionization mass spectroscopy (EDB-MS) (Birdsall et al., 2018). This will
lead to a step change in our understanding of how such particles evolve in the atmosphere by directly linking optical properties to chemical composition.

*Data availability.* For data related to this paper contact Nir Bluvshtein (nir.bluvshtein@env.ethz.ch) or Ulrich K. Krieger (Ulrich.krieger@env.ethz.ch).

*Acknowledgements.* NB is grateful for support from the ETH Zurich Postdoctoral Fellowship program.

*Competing interests.* The authors declare that they have no conflict of interest.

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

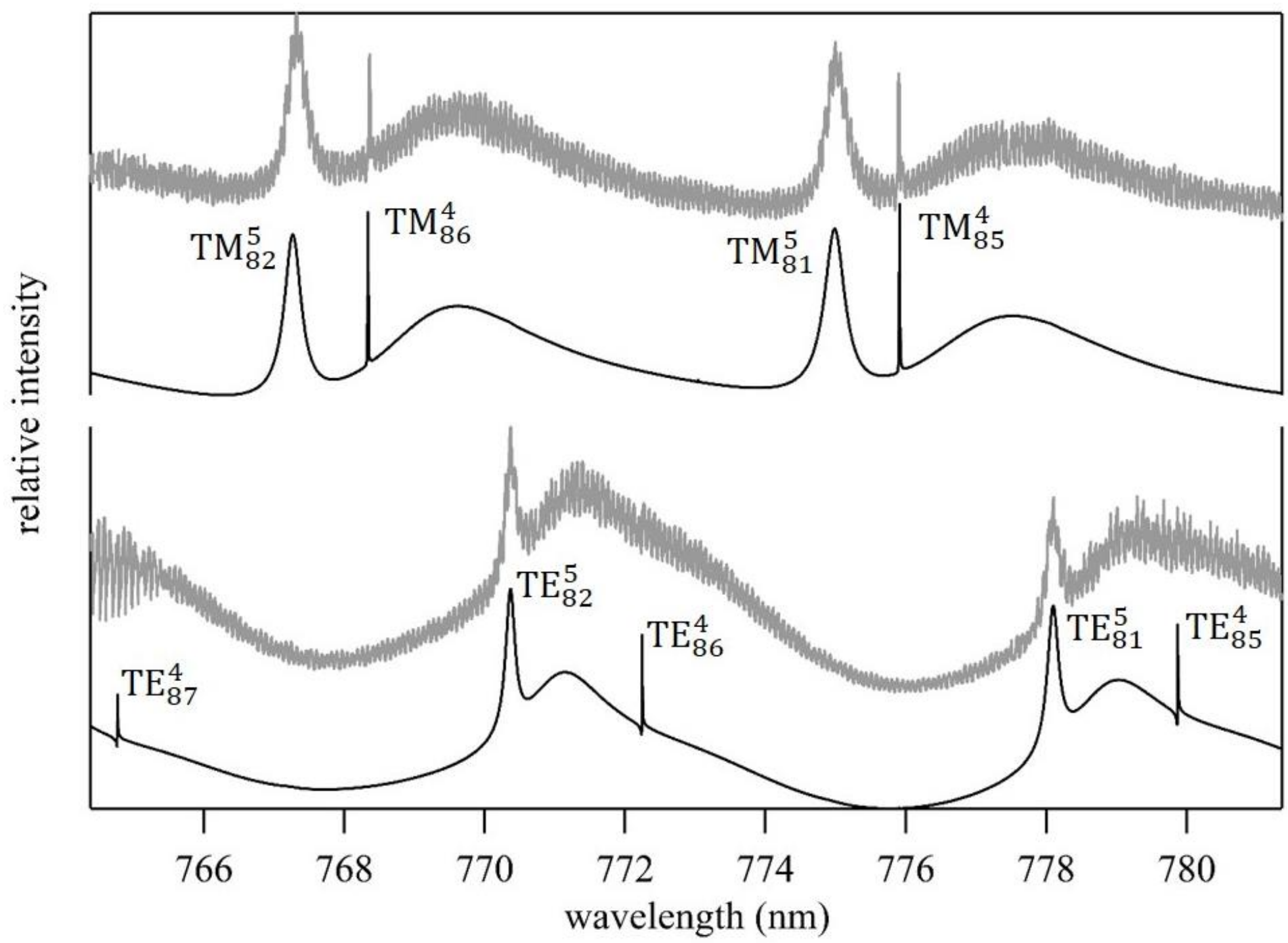

**Figure 3. Measured Mie scattering spectra at $\pi$/2 angle of a levitated particle. The positions of the TE and TM resonance modes (e.g. $TM_n^l$; identified by their order ($l$) and mode ($n$) numbers) were used to retrieve the particle's real refractive index ($n_D$ =1.4665, $m_1$ =2745) and size ($a$ = 9.2906 µm). We note that the lower order modes in the fitted TE spectrum are too narrow to be observed in the measured spectrum with the current resolution.**
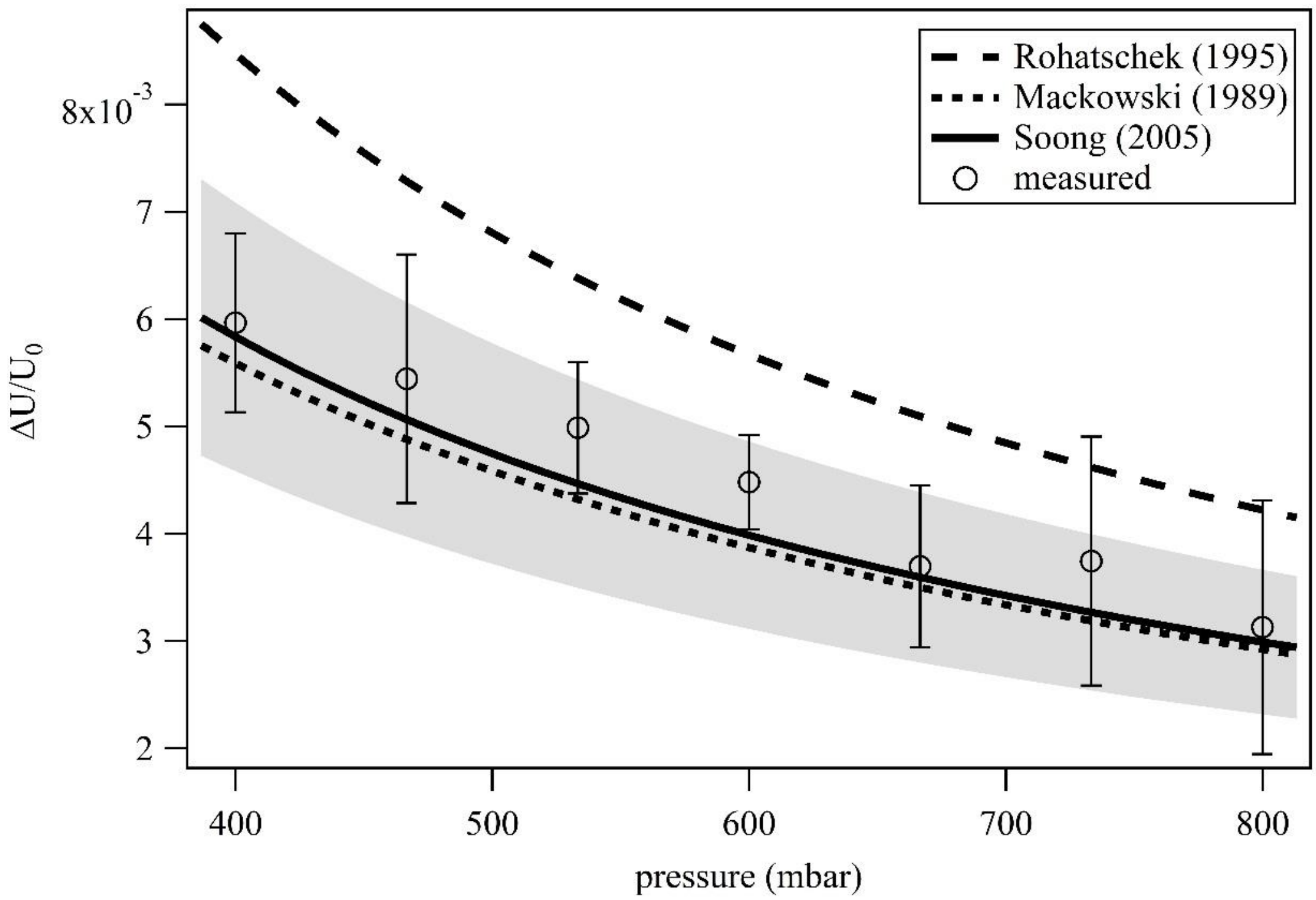

545 **Figure 4. Measured and simulated EDB response ($\Delta U/U_0$) to the illumination of a slightly absorbing particle due to the photophoretic effect. The two analytical solutions; Mackowski (1989) dotted line and Soong et al. (2005) solid line, that account for slip-flow conditions agree well with our measurements (empty circles). Error bars: standard deviation over five illumination cycles. Gray shaded area: measurement uncertainty propagated through the Soong model calculation, dominated by the 15% uncertainty on the radiant flux measured with a power meter and a beam profiler.**

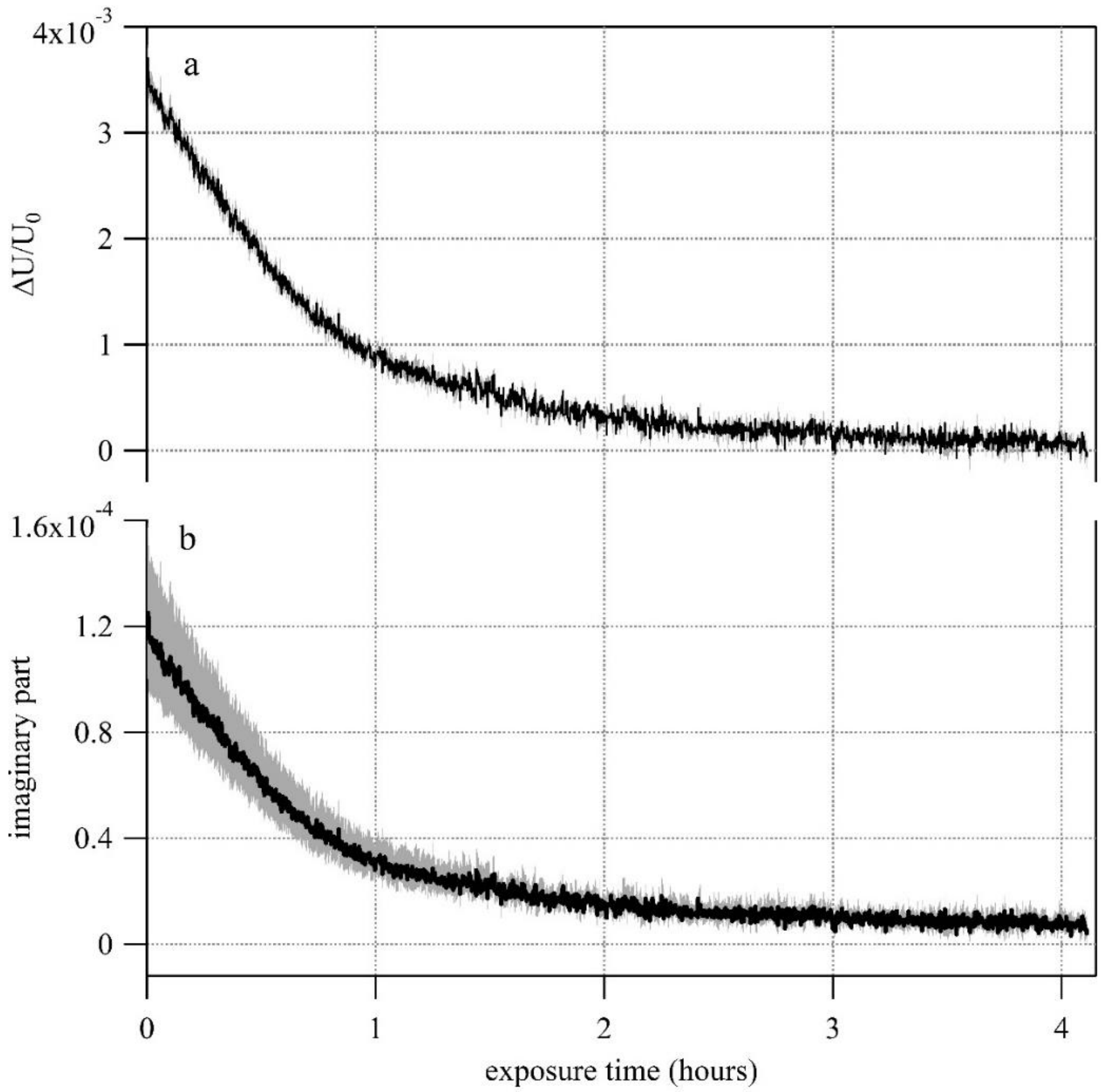

550 **Figure 5. (a) Decay of the EDB photophoretic response ($\Delta U/U_0$) during illumination of a slightly absorbing particle (PEG400 with 0.23% wt of carminic acid; CA). The decay is due to the slow photolysis of the CA. (b) The retrieved imaginary part of the complex refractive index. Gray shaded area in both plots represents the uncertainty in the measured and retrieved values.**