# Peer review of "Photophoretic spectroscopy in atmospheric chemistry – high sensitivity measurements of light absorption by a single particle"

_Atmospheric Measurement Techniques, 2020_

## Referee Comment (RC1) · Anonymous Referee #1 · 18 Mar 2020

The authors report a photophoretic technique as applied to levitated droplets in a double-ring electrodynamic balance. They show that the droplet position changes slightly when slightly light-absorbing molecules are present in the sample, due to the indirect photophoretic effect. By relating the change in DC balance voltage to the change in position, the authors are able to determine the complex part of the refractive index (cRI). The authors demonstrate the retrieval of the cRI of carminic acid in a PEG400 solution at 0.23wt% and report agreement within error to their bulk measurements.

This is a nice new technique and demonstrates another method that the cRI can be established from measurements on levitated droplets. However, it appears to me that

the application of this method to light-absorbing properties of atmospheric sample may be limited. Overall, the paper is well-written and falls within the scope of the journal. Clear answers to the following questions/comments would alleviate my concerns over the applicability of this technique:

1. What range of k is this method appropriate for? An upper and lower limit should be estimated to allow readers to gauge the effectiveness of the method in characterizing brown carbon samples.

2. Would this method be equally applicable to molecules that absorb less strongly but that are present at higher concentration?

3. What range of samples/conditions (particularly RH) can this method be applied to? I would assume that any water vapor in equilibrium with the droplet would be affected by light-absorption and thus add additional complexity to the interpretation of the droplet movement with the laser illumination.

4. Does this measurement require the sample be dissolved in a solution with PEG400 (or similar) as a solvent? How could this technique be expanded outside of this solvent system given the need for well-characterized thermal accommodation coefficients and thermal conductivity?

5. How does $\Delta U/U\_0$ vary with a non-absorbing sample (pure PEG400 for example)? This is a necessary benchmark to show that there is no measurable effect when no absorption occurs.

6. The main reason to make measurements on levitated droplets rather than bulk samples is that the super-saturated and super-cooled states accessible to aerosol in the atmosphere can be reproduced in a well-controlled lab environment. I would like the authors expand on the applicability of their method to these kinds of samples.

7. What causes the oscillation in the Mie resonance spectra compared to the simulation? Is this from the laser or related to the movement of the droplet with the heating

beam?

8. Please show a figure (or SI figure) of the DC voltage as a function of time as the heating laser cycles on/off.

Thank you for presenting a very interesting new method!

---

## Referee Comment (RC2) · Anonymous Referee #2 · 21 Mar 2020

Questions:
This seems to be only applicable to droplets that are neither strongly nor weakly absorbing (i.e. the $k$ range of $10^{-4}$ to to $10^{-5}$ stated in the abstract).
a. Is that the actual dynamic range?
b. Isn't $k$ for brown carbon (BrC) typically much larger than those values? I mention this because BrC is presented as the motivation for the development of the technique.
c. What atmospheric systems could this technique be applied to? I would imagine weak absorbers (e.g. aqueous sea salt aerosol) have a $k$ that is too small while strong absorbers (the aforementioned BrC) have a $k$ that is too large.
d. Why no measurements of, for example, aqueous humic acid?

Minor comments:
-no need to use a non-standard symbol like $\chi$ for the size parameter when the standard $x$ will do fine.
-Line 134: "retrieved by minimizing the difference between measured and calculated wavelength" I assume you mean minimizing the sum of squared differences?
-use of the times symbol in many equations is unnecessary.
-Line 225: What is the viscosity of PEG400?
-Line 269: "473-nm" to "473 nm"
-Figure 2 caption: intensity units are italics when they should not be.

---

## Author Comment (AC1) · 15 Apr 2020

We would like to thank Anonymous Referee #1 for carefully reading our submitted manuscript and for raising important and insightful questions that helped us improve the quality and readability of the manuscript.

answer to question #1:

There are no theoretical upper and lower limits for retrievals of $\kappa$ using the described method. On the lower end, this would depend on the sensitivity and stability of the experimental set-up. With our system, which could be improved, as mentioned in the

manuscript this value is about 10ˆ-5. On the upper end of $\kappa$ values, more energy is absorbed and as a result, the signal is generally (not always, see below) stronger and easier to measure. However, this should be compensated with reducing the power of the light source to avoid significant temperature change of the particle.

We chose to emphasize the usefulness of EDB-PPS at the lower range of $\kappa$. Because at higher values of $\kappa$, other particle-phase techniques are available, that are simpler to implement. These include photoacoustic spectroscopy, cavity-enhanced spectroscopy, extinction minus scattering, and filter based techniques. Therefore one can expect this method to be mostly rewarding at $\kappa \leq$ 10ˆ-3 in aging experiments that are intended to follow 'bleaching' or 'browning' of BrC.

To better explain the limitations of this technique with regards to a useful range of the imaginary part we added the following figure (figure 1 below). This figure illustrates the simulated system signal ($\Delta$U/U0) over 6 orders of magnitude of $\kappa$, from 10ˆ-8 to 10ˆ-2, of a 10 $\mu$m, PEG400 based particle. At the lower end of this range (left side of figure 1a), $\kappa$ is effectivity zero and the simulated signal is negative (particle moves away from the light source) due to direct photophoresis (radiation pressure) alone. Note how the signal increases to positive values as $\kappa$ increases (figure 1a and 1b) due to an increase in the thermal asymmetry parameter. With the increase in $\kappa$ there more energy that is absorbed on the illuminated side of the particle. This offsets the hotspot of absorbed energy at the 'dark' side of the particle that is due to 'nano-focusing'. As a result, the thermal asymmetry parameter is reduced together with the signal. As $\kappa$ continues to increase the asymmetry parameter changes sign from positive to negative as the illuminated side of the particle becomes warmer than the 'dark' side. This is shown in figure 1c as a negative signal and illustrated in 1d as the particle shifts from the regime described by the upper part to the one in the lower part of figure 1d.

Based on this description and on our system's sensitivity limitations we determined the lower limit for $\kappa$ retrieval at 10ˆ-5 and do not determine an upper limit. We do, however, note areas with high uncertainty, namely, around $\kappa \approx$ 10ˆ-3, where the change in signal

flattens.

An important caveat is a non-injective behavior in figure 1b. To address the issue one would need to understand the order of magnitude of the particle's absorption (for example; is it below or above $\kappa = 10^{-3}$). Additionally, in aging experiments, as $\kappa$ evolves, the direction of the change of the signal would clarify the direction of the change of $\kappa$.

This figure will be added as figure #7 and a descriptive text will be added to the conclusion section in the revised version of this manuscript to describe the figure and the simulated system behavior in different absorptivity regimes.

answer to question #2:

The retrieved refractive index or the imaginary part of the refractive index, like in other retrieval methods, is that which best represent the optical properties of the measured particle. For a single composite particle that would be the refractive index of the bulk material. For a multi-component particle (miscible liquids, solutions, etc.) the retrieved refractive index sometimes referred to as an effective refractive index could be different from that of the individual materials. In the results given in this manuscript, the retrieved imaginary part is not that of carminic acid and not that of PEG400. It is the imaginary part of the specific mixture and is a product of the amount of light absorbed by the particle. A different type of absorbing molecules i.e. with higher or lower molar absorptivity at higher or lower concentration would yield a different imaginary part based on the amount of light absorbed by the mixture. In this respect, this method is not different from any other retrieval method in bulk, thin film or in the particle phase.

answer to question #3:

Illuminating light-absorbing particles results in inevitable temperature change that is related to the amount of light absorbed and to the materials heat capacity. For the particles in this study, the temperature change is estimated to be below 1 K by using

[Figure]

the heat conductivity equation (from 0.02 to 0.4 K, depending on $\kappa$ and on the size of the particle). In equilibrium, this would also depend on the heat conductivity of the particle and its surrounding and the surrounding temperature. A hydrated particle, upon increasing temperature, will lose some water and will reach a new equilibrium with its environment at lower water activity even though the RH remains unchanged. This also means that at equilibrium the particle will be smaller. The extent of this effect depends on the temperature change within the particle. This effect was used as a method to retrieve $\kappa$ (Knox & Reid, 2008; Willoughby et al., 2017). There is no theoretical limitation for retrieval of $\kappa$ using EDB-PPS at elevated RH conditions. Temperature equilibrium is reached within milliseconds while water activity and size equilibrium are reached depending on the particle diffusivity. In the case of PEG400, this happens within a time scale of about 1 second. As long as equilibrium is reached, the signal can be used. The added complexity compared to a dry experiment is that parameters such as thermal conductivity of the particle and of the environment as well as the size of the particle (before and during illumination) are needed. We have performed several experiments with the PEG400-CA system at elevated RH. In these experiments, the size of the particle before illumination is retrieved with the high resolution scattering spectra as described in the manuscript, while the instantaneous size change due to water lose/gain was monitored with using low resolution scattering spectra at 1 Hz at a resolution of about 1 nm (Steimer et al., 2015). It was clear that size (and water activity) equilibrium was reached within 1 second/spectrum. Further exploring this application was beyond the scope of this manuscript but will most likely be a part of future implementation of the EDB-PPS.

Knox, K. J., & Reid, J. P. (2008). Ultrasensitive absorption spectroscopy of optically-trapped aerosol droplets. Journal of Physical Chemistry A, 112(42), 10439–10441. https://doi.org/10.1021/jp807418g

Steimer, S. S., Krieger, U. K., Te, Y. F., Lienhard, D. M., Huisman, A. J., Luo, B. P., Ammann, M., & Peter, T. (2015). Electrodynamic balance measurements of thermodynamic, kinetic, and optical aerosol properties inaccessible to bulk methods. Atmos. Meas. Tech., 8(6), 2397–2408. https://doi.org/10.5194/amt-8-2397-2015

Willoughby, R. E., Cotterell, M. I., Lin, H., Orr-Ewing, A. J., & Reid, J. P. (2017). Measurements of the Imaginary Component of the Refractive Index of Weakly Absorbing Single Aerosol Particles. Journal of Physical Chemistry A, 121(30), 5700–5710. https://doi.org/10.1021/acs.jpca.7b05418

answer to question #4:

This technique does not require the sample to be dissolved in PEG400 or any other solvent. The goal of this work is to retrieve low levels of $\kappa$; in a range, other particle-phase techniques are not sensitive enough. For this reason, PEG400 was used as a non-absorbing organic matrix that allows for the total absorptivity of the particle to be much lower compared to pure carminic acid. Nevertheless, as the reviewer pointed out, a disadvantage of this method is the need for some parameters to be well characterized. Amongst others, these include thermal conductivity (of both particle and gas phase) and thermal accommodation coefficient.

Thermal conductivity: uncertainty in thermal conductivity leads to uncertainty in the retrieved $\kappa$. As an example, an uncertainty of 30% in the thermal conductivity leads to a non-symmetric 20 − 30% uncertainty in retrieved $\kappa$ in the range of 10-5 – 10-3. The range of uncertainty is due to the non-linearity of $\kappa$ with $\Delta U/U0$.

One way to reduce the uncertainty is to perform measurements of an unknown absorbing material which is highly diluted or well mixed with a well-characterized material (such as PEG400, water, sulfuric acid). In this way, the thermal properties of the particle are assumed identical to the thermal properties of the well-characterized material.

An additional approach for unknown, atmospherically relevant organic material is to consider that many organic compounds are very similar with respect to their thermal properties and use an approximated value with an appropriate uncertainty. We refer the

reviewer to the literature survey by Latini et al (2014) that compiled thermal conductivity data for the following 3400 data sets (excluding additional 1340 data sets of refrigerant compounds) at atmospheric pressure and reduced temperature of about $0.6 \pm 0.14$.

Thermal accommodation coefficient ($\alpha$T): the literature on measurements or estimations of $\alpha$T is scarce. As a result, one must simply assume a reasonable value. It is important to note that an error in $\alpha$T propagates to relatively small errors in the retrieved $\kappa$. For example, the value that we used in this study i.e. $\alpha$T=0.85 $\pm$0.15 ($\pm\sim$18%) leads to a non-symmetric $5 - 7\%$ uncertainty in retrieved $\kappa$ in the range of $10^{-5} - 10^{-3}$.

This discussion will be added to the last paragraph in section 3 of the revised manuscript.

Latini, G., Di Nicola, G., & Pierantozzi, M. (2014). A critical survey of thermal conductivity literature data for organic compounds at atmospheric pressure and an equation for aromatic compounds. Energy Procedia, 45, 616–625. https://doi.org/10.1016/j.egypro.2014.01.066

answer to question #5:

For non-absorbing particles, the only effect from illuminating it would be that of the direct photophoresis (radiation pressure), which is about $1 - 2$ orders of magnitude smaller than indirect photophoresis in our experimental range. Its direction is always away from the light source so if detectable, it should result in a negative signal (i.e. negative $\Delta$U/U0).

An experiment with undyed PEG400 particle showed no detectable signal (i.e. above noise level). The following text will be added to section 3: "It is important to note that as expected, for a pure PEG400 particle (i.e. no measurable absorption at 473 nm) at similar conditions, but a slightly less sensitive setup, no signal could be detected."

answer to question #6:

We agree with the reviewer and mentioned this in the text, that a significant disadvantage of bulk and film type of experiments is that supersaturated conditions are not accessible.

This point will be made clearer at the end of the introduction section of the revised version of the manuscript.

answer to question #7:

If the reviewer is referring to figure 3, this figure shows the resonance peaks which are labeled on the theoretical curves and additional noise oscillations that extend the full spectra in both polarizations. These oscillations are caused mainly by variations in the laser power as the length of its optical cavity is changing. The particle horizontal movement due to the AC field sometimes causes an additional noise component. Most (but not all) of the first component is reduced by dividing the particle spectrum by a reference spectrum. Note that in this study, the particle was not illuminated with the "heating" beam while scanned with the TDL laser.

The following text will be added to the figure caption: "Residual noise in the measured spectrum originates mostly from laser power oscillations due to frequency scanning operation and from the horizontal oscillation of the particle due to the applied AC field."

answer to question #8:

The following figure 2 will be added to the revised manuscript.

[Figure]

**Fig. 1.** simulated signal (and uncertainty) for a PEG400 particle, 10 $\mu$m in radius at T = 20 C, P = 400 mbar, m1=1.466, $\lambda$=473 nm, I=35 mW mm^-2 and full thermal accommodation.

**Fig. 2.** change in the applied DC voltage in response to alternately illuminating a trapped light absorbing particle. Here, a particle with 12.72 $\mu$m radius and $\kappa \approx 1.36 \times 10\hat{}$-4.

---

## Author Comment (AC2) · 15 Apr 2020

We would like to thank anonymous referee #2 as well, for carefully reading our submitted manuscript and for raising important questions that helped us improve the quality and readability of the manuscript.

Answer to question a:

Please refer to the answer given to question #1 from referee #1. Figure 1 in the reply to referee #1 will be added to address this question.

Answer to question b:

Values of  $\kappa$  for BrC reported in the literature are limited by the detection limit and sensitivity of the measurement technique that is used. Therefore, a more accurate statement would be that quantifiable values of  $\kappa$  for BrC are typically much larger than 10-4 while the reality is that there is no physical lower limit for absorptivity of BrC in the atmosphere.

An alternative argument could be made that lower values of  $\kappa$  for aerosols are insignificant due to their low thermal effect (at least in the troposphere). It is, therefore, worth clarifying that the main motivation for the application of this technique is to measure these previously unquantifiable values and to follow the evolution processes of BrC during atmospheric aging. This could mean either 'bleaching' or 'browning' and for that reason, one should be able to retrieve  $\kappa$  at lower values that are commonly reported.

To answer this comment and comment number 6 from reviewer 1 we will add the following text at the end of the introduction in the revised version of the manuscript:

"With this approach, we gain from combining the advantage of light absorption sensitivity nearing that of bulk UV-vis measurements with the advantage of studying chemical processes of the particle phase (accessible supersaturated conditions) in an environmental chamber able to simulate a wide range of atmospheric conditions. This could contribute to the study of light absorption evolution during atmospheric aging of BrC aerosols."

Answer to question c:

As mentioned above, any color-forming or degrading aging processes could be relevant candidates. Oligomerization, nitrification, acid-catalyzed dehydration have been reported as a possible mechanism for chromophore formation. Oxidation and photooxidation reactions are known bleaching processes. One particular system of interest is the browning of sulfuric acid aerosols in the presence of ketones or aldehydes. For this system, the formation of UV-Vis chromophores was reported so far only in bulk and film experiments but not for aerosols.
Answer to question d:

We appreciate the reviewer's suggestion and agree that such a measurement would be beneficial to demonstrate the usefulness of this method. To produce a sample of humic or fulvic acid material with low absorptivity in the range of  $\kappa \leq 10^{-3}$  at 473 nm it would need to be highly diluted or mixed with a non-absorbing material. This cannot be achieved with water because these particles are not sufficiently hygroscopic (additionally, we refer the reviewer to a short discussion about measurements at high RH in the answer to question 3 by reviewer #1). Unfortunately, this was attempted but could also not be achieved with PEG400 due to extremely low solubility. An additional option would be to simply use a longer wavelength for the excitation light source. This was, however, not available at the time of performing these experiments.

Minor comments:

-no need to use a non-standard symbol like  $\chi$  for the size parameter when the standard x will do fine.

The notation for size parameter was changed from " $\chi$ " to "x" in the text as well as in all of the equations.

-Line 134: "retrieved by minimizing the difference between measured and calculated wavelength" I assume you mean minimizing the sum of squared differences?

What was actually done was to minimize the sum of the absolute value of the differences. This will be added in the text.

-use of the times symbol in many equations is unnecessary.

To improve clarity, the multiplication symbol was removed from all equations.

-Line 225: What is the viscosity of PEG400?

A range of viscosity values for PEG400 as reported by the supplier (Merck), at 293.15 K was added to the text; 105 - 130 mPa sec.
-Line 269: "473-nm" to "473 nm"

This was correct in the text.

-Figure 2 caption: intensity units are italics when they should not be

This was corrected in the text.

---

## Author Response (AR1)

We would like to thank the reviewers for carefully reading our submitted manuscript and for raising important and insightful questions that helped us improve the quality and readability of the manuscript.

In what fallow below, the comments made by the reviewers are in black and our answers are in red. All changes to the manuscript are marked by "track changes" in the manuscript file.

**Anonymous Referee #1**

The authors report a photophoretic technique as applied to levitated droplets in a double-ring electrodynamic balance. They show that the droplet position changes slightly when slightly light-absorbing molecules are present in the sample, due to the indirect photophoretic effect. By relating the change in DC balance voltage to the change in position, the authors are able to determine the complex part of the refractive index (cRI). The authors demonstrate the retrieval of the cRI of carminic acid in a PEG400 solution at 0.23 wt% and report agreement within error to their bulk measurements. This is a nice new technique and demonstrates another method that the cRI can be established from measurements on levitated droplets. However, it appears to me that the application of this method to light-absorbing properties of atmospheric sample may be limited. Overall, the paper is well-written and falls within the scope of the journal. Clear answers to the following questions/comments would alleviate my concerns over the applicability of this technique:

1. What range of k is this method appropriate for? An upper and lower limit should be estimated to allow readers to gauge the effectiveness of the method in characterizing brown carbon samples.

There are no theoretical upper and lower limits for retrievals of $\kappa$ using the described method. On the lower end this would depend on the sensitivity and stability of the experimental set-up. With our system, which could be improved, as mentioned in the manuscript this value is about $10^{-5}$. On the upper end of $\kappa$ values, more energy is absorbed and as a result the signal is generally (not always, see below) stronger and easier to measure. However, this should be compensated with reducing the power of the light source to avoid significant temperature change of the particle.

We chose to emphasize the usefulness of EDB-PPS at the lower range of $\kappa$. Because at higher values of $\kappa$, other particle phase techniques are available, that are simpler to implement. These include photoacoustic spectroscopy, cavity enhanced spectroscopy, extinction minus scattering and filter based techniques. Therefore one can expect this method to be mostly rewarding at $\kappa \leq 10^{-3}$ in aging experiments that are intended to follow 'bleaching' or 'browning' of BrC.

To better explain the limitations of this technique with regards to a useful range of the imaginary part we added the following figure (figure 7 in the revised manuscript). This figure illustrates the simulated system signal ($\Delta U/U_0$) over 6 orders of magnitude of $\kappa$, from $10^{-8}$ to $10^{-2}$, of a 10 μm, PEG400 based particle. At the lower end of this range (left side of figure 7a) $\kappa$ is effectivity zero and the simulated signal is negative (particle moves away from the light source) due to direct photophoresis (radiation pressure) alone. Note how the signal increases to positive values as $\kappa$ increases (figure 7a and 7b) due to an increase in the thermal asymmetry parameter. With the increase in $\kappa$ there more energy that is absorbed on the illuminated side of the particle. This offsets the hotspot of absorbed energy at the 'dark' side of the particle that is due to 'nano-focusing'. As a result, the thermal asymmetry parameter is reduced together with the signal. As

κ continues to increase the asymmetry parameter changes sign from positive to negative as the illuminated side of the particle becomes warmer than the 'dark' side. This is shown in figure 7c as a negative signal and illustrated in 7d as the particle shifts from the regime described by the upper part to the one in the lower part of figure 7d.

Based on this description and on our system's sensitivity limitations we determined the lower limit for κ retrieval at $10^{-5}$ and do not determine an upper limit. We do however, note areas with high uncertainty, namely, around $\kappa \approx 10^{-3}$, where the change in signal flattens.

An important caveat is the non-injective behavior in figure 7b. To address the issue one would need to understand the order of magnitude of the particle's absorption (for example; is it below or above $\kappa = 10^{-3}$). Additionally, in aging experiments, as κ evolves, the direction of the change of the signal would clarify the direction of the change of κ.

[Figure]

Descriptive text was added to the conclusion section in the revised version of this manuscript to describe the figure and the simulated system behavior in different absorptivity regimes.

2. Would this method be equally applicable to molecules that absorb less strongly but that are present at higher concentration?

The retrieved refractive index or the imaginary part of the refractive index, like in other retrieval methods, is that which best represent the optical properties of the measured particle. For a single composite particle that would be the refractive index of the bulk material. For a multi

component particle (miscible liquids, solutions etc.) the retrieved refractive index sometimes referred to as an effective refractive index could be different from that of the individual materials. In the results given in this manuscript, the retrieved imaginary part is not that of carminic acid and not that of PEG400. It is the imaginary part of the specific mixture and is a product of the amount of light absorbed by the particle. A different type of absorbing molecules i.e. with higher or lower molar absorptivity at higher or lower concentration would yield a different imaginary part based on the amount of light absorbed by the mixture. In this respect, this method is not different from any other retrieval method in bulk, thin film or in particle phase.

3. What range of samples/conditions (particularly RH) can this method be applied to? I would assume that any water vapor in equilibrium with the droplet would be affected by light-absorption and thus add additional complexity to the interpretation of the droplet movement with the laser illumination.

Illuminating light absorbing particle results in inevitable temperature change that is related to the amount of light absorbed and to the materials heat capacity. For the particles in this study, the temperature change is estimated to be below 1 K with using the heat conductivity equation (from 0.02 to 0.4 K, depending on κ and on the size of the particle). In equilibrium, this would also depend on the heat conductivity of the particle and its surrounding, and the surrounding temperature. A hydrated particle, upon increasing temperature, will lose some water and will reach new equilibrium with its environment at lower water activity even though the RH remains unchanged. This also means that at equilibrium the particle will be smaller. The extant of this effect depends of the temperature change within the particle. This effect was used as a method to retrieve κ (Knox & Reid, 2008; Willoughby et al., 2017). There is no theoretical limitation for retrieval of κ using EDB-PPS at elevated RH conditions. Temperature equilibrium is reached within milliseconds while water activity and size equilibrium is reached depending on the particles diffusivity. In the case of PEG400 this happens within a time scale of about 1 second. As long as equilibrium is reached, the signal can be used. The added complexity compared to a dry experiment is that parameters such as thermal conductivity of the particle and of the environment as well as the size of the particle (before and during illumination) are needed. We have performed several experiment with the PEG400-CA system at elevated RH. In these experiments, the size of the particle before illumination is retrieved with the high resolution scattering spectra as described in the manuscript, while the instantaneous size change due to water lose/gain was monitored with using low resolution scattering spectra at 1 Hz at resolution of about 1 nm (Steimer et al., 2015). It was clear that size (and water activity) equilibrium was reached within 1 second/spectrum. Further exploring this application was beyond the scope of this manuscript but will most likely be a part of future implementation of the EDB-PPS.

4. Does this measurement require the sample be dissolved in a solution with PEG400 (or similar) as a solvent? How could this technique be expanded outside of this solvent system given the need for well-characterized thermal accommodation coefficients and thermal conductivity?

This technique does not require the sample to be dissolved in PEG400 or any other solvent. The goal of this work is to retrieve low levels of κ; in a range, other particle phase techniques are

not sensitive enough. For this reason, PEG400 was used as a non-absorbing organic matrix that allows for the total absorptivity of the particle to be much lower compared to pure carminic acid. Nevertheless, as the reviewer pointed out, a disadvantage of this method is the need for some parameters to be well characterized. Amongst others these include thermal conductivity (of both particle and gas phase) and thermal accommodation coefficient.

Thermal conductivity: uncertainty in thermal conductivity leads to uncertainty in the retrieved κ. As an example, uncertainty of 30% in the thermal conductivity leads to a non-symmetric 20 – 30% uncertainty in retrieved κ in the range of $10^{-5} – 10^{-3}$. The range of uncertainty is due to non-linearity of κ with $\Delta U/U_0$.

One way to reduce the uncertainty is to perform measurements of an unknown absorbing material which is highly diluted or well mixed with a well characterized material (such as PEG400, water, sulfuric acid). In this way, the thermal properties of the particle are assumed identical to the thermal properties of the well-characterized material.

An additional approach for unknown, atmospherically relevant organic material is to consider that many organic compounds are very similar with respect to their thermal properties and use an approximated value with an appropriate uncertainty. We refer the reviewer to the literature survey by Latini et al (2014) that compiled thermal conductivity data for the following 3400 data sets (excluding additional 1340 data sets of refrigerant compounds) at atmospheric pressure and reduced temperature of about $0.6 \pm 0.14$:

|  | N | mean | SD | %SD |
|---|---|---|---|---|
| Alcohols | 775 | 0.1482 | 0.0256 | 17.3 |
| Alkanes | 1025 | 0.1259 | 0.0182 | 14.5 |
| Alkenes | 135 | 0.1301 | 0.0244 | 18.7 |
| Aromatics | 570 | 0.1174 | 0.0180 | 15.3 |
| Carboxylic acids | 318 | 0.1576 | 0.0427 | 27.1 |
| Cycloalkanes | 35 | 0.1262 | 0.0134 | 10.6 |
| Cycloalkenes | 10 | 0.1303 | 0.0104 | 8.0 |
| Esters | 236 | 0.1261 | 0.0179 | 14.2 |
| Ethers | 111 | 0.1268 | 0.0181 | 14.3 |
| Ketones | 185 | 0.1417 | 0.0203 | 14.3 |

Thermal accommodation coefficient ($\alpha_T$): the literature on measurements or estimations of $\alpha_T$ is scarce. As a result, one must simply assume a reasonable value. It is important to note that an error in $\alpha_T$ propagate to relatively small errors in the retrieved κ. For example, the value that we used in this study i.e. $\alpha_T = 0.85 \pm 0.15 (\pm\sim18\%)$ leads to a non-symmetric 5 – 7% uncertainty in retrieved κ in the range of $10^{-5} – 10^{-3}$.

This discussion was added to the last paragraph in section 3 of the revised manuscript.

5. How does ΔU/U_0 vary with a non-absorbing sample (pure PEG400 for example)? This is a necessary benchmark to show that there is no measurable effect when no absorption occurs.

For non-absorbing particle, the only effect from illuminating it would be that of the direct photophoresis (radiation pressure), which is about 1 – 2 orders of magnitude smaller than

indirect photophoresis in our experimental range. Its direction is always away from the light source so if detectable, it should result in a negative signal (i.e. negative $\Delta U/U_0$).

An experiment with undyed PEG400 particle showed no detectable signal (i.e. above noise level).

The following text was added to section 3: "It is important to note that as expected, for a pure PEG400 particle (i.e. no measurable absorption at 473 nm) at similar conditions, but a slightly less sensitive setup, no signal could be detected."

6. The main reason to make measurements on levitated droplets rather than bulk samples is that the super-saturated and super-cooled states accessible to aerosol in the atmosphere can be reproduced in a well-controlled lab environment. I would like the authors expand on the applicability of their method to these kinds of samples.

We agree with the reviewer, and mentioned in the text, that a significant disadvantage of bulk and film type of experiments is that super saturated conditions are not accessible.

This point was made clearer at the end of the introduction section of the revised version of the manuscript.

7. What causes the oscillation in the Mie resonance spectra compared to the simulation? Is this from the laser or related to the movement of the droplet with the heating beam?

If the reviewer is referring to figure 4 (previously figure 3), this figure shows the resonance peaks which are labeled on the theoretical curves and additional noise oscillations that extend the full spectra in both polarizations. These oscillations are caused mainly from variations in the laser power as the length of its optical cavity is changing. The particle horizontal movement due to the AC field sometimes causes an additional noise component. Most (but not all) of the first component is reduced by dividing the particle spectrum by a reference spectrum. Note that in this study, the particle was not illuminated with the "heating" beam while scanned with the TDL laser.

The following text was added to the figure caption: "Residual noise in the measured spectrum originate mostly from laser power oscillations due to frequency scanning operation and from horizontal oscillation of the particle due to the applied AC field."

8. Please show a figure (or SI figure) of the DC voltage as a function of time as the heating laser cycles on/off.

The following figure was added as figure 2 in the revised manuscript.

[Figure]

**Anonymous Referee #2**

Questions:
This seems to be only applicable to droplets that are neither strongly nor weakly absorbing (i.e. the $k$ range of $10^{-4}$ to to $10^{-5}$ stated in the abstract).

a. Is that the actual dynamic range?

Please refer to the answer given to question #1 from referee #1. Figure 7 in the revised version was added to address this question.

b. Isn't $k$ for brown carbon (BrC) typically much larger than those values? I mention this because BrC is presented as the motivation for the development of the technique.

Values of $\kappa$ for BrC reported in the literature are limited by the detection limit and sensitivity of the measurement technique that is used. Therefore, a more accurate statement would be that quantifiable values of $\kappa$ for BrC are typically much larger than $10^{-4}$ while the reality is that there is no physical lower limit for absorptivity of BrC in the atmosphere.

An alternative argument could be made that lower values of $\kappa$ for aerosols are insignificant due to their low thermal effect (at least in the troposphere). It is, therefore, worth clarifying that the main motivation for the application of this technique is to measure these previously unquantifiable values and to follow evolution processes of BrC during atmospheric aging. This could mean either 'bleaching' or 'browning' and for that reason one should be able to retrieve $\kappa$ at lower values that are commonly reported.

To answer this comment and comment number 6 from reviewer 1 we add the following text at the end of the introduction in the revised version of the manuscript: "With this approach we gain from combining the advantage of light absorption sensitivity nearing that of bulk UV-vis measurements with the advantage of studying chemical processes of the particle phase (accessible super saturated conditions) in an environmental chamber able to simulate a wide

range of atmospheric conditions. This could contribute to the study of light absorption evolution during atmospheric aging of BrC aerosols."

c. What atmospheric systems could this technique be applied to? I would imagine weak absorbers (e.g. aqueous sea salt aerosol) have a $k$ that is too small while strong absorbers (the aforementioned BrC) have a $k$ that is too large.

As mentioned above, any color forming or degrading aging processes could be relevant candidates. Oligomerization, nitrification, acid catalyzed dehydration have been reported as possible mechanism for chromophore formation. Oxidation and photooxidation reactions are known bleaching processes. One particular system of interest is browning of sulfuric acid aerosols in the presence of ketones or aldehydes. For this system, formation of UV-Vis chromophores was reported so far only in bulk and film experiment but not for aerosols.

d. Why no measurements of, for example, aqueous humic acid?

We appreciate the reviewer suggestion and agree that such a measurement would be beneficial to demonstrate the usefulness of this method. To produce a sample of humic or fulvic acid material with low absorptivity in the range of $\kappa \leq 10^{-3}$ at 473 nm it would need to be highly diluted or mixed with a non-absorbing material. This cannot be achieved with water because these particles are not sufficiently hygroscopic (additionally, we refer the reviewer to a short discussion about measurements at high RH in the answer to question 3 by reviewer #1). Unfortunately, this was attempted but could also not be achieved with PEG400 due to extremely low solubility. An additional option would be to simply use a longer wavelength for the excitation light source. This was, however, not available at the time of performing these experiments.

Minor comments:

-no need to use a non-standard symbol like $\chi$ for the size parameter when the standard $x$ will do fine.

The notation for size parameter was changed from "$\chi$" to "$x$" in the text as well as in all of the equations.

-Line 134: "retrieved by minimizing the difference between measured and calculated wavelength" I assume you mean minimizing the sum of squared differences?

What was actually done was to minimize the sum of the absolute value of the differences. This was added in the text.

-use of the times symbol in many equations is unnecessary.

To improve clarity, the multiplication symbol was removed from all equations.

-Line 225: What is the viscosity of PEG400?

A range of viscosity values for PEG400 as reported by the supplier (Merck), at 293.15 K was added to the text; 105 - 130 mPa sec.

-Line 269: "473-nm" to "473 nm"

This was correct in the text.

-Figure 2 caption: intensity units are italics when they should not be

This was corrected in the text.

Additional changes in the revised version of the manuscript:

1) In figure 5 (previously figure 4), the units of the X axis were changed from mbar to hPa to match the units used in the text.
2) The Y axis of figure 6 (previously figure 5) was changed to log scale.
3) Several typos were identified and corrected.

Knox, K. J., & Reid, J. P. (2008). Ultrasensitive absorption spectroscopy of optically-trapped aerosol droplets. *Journal of Physical Chemistry A*, *112*(42), 10439–10441. https://doi.org/10.1021/jp807418g

Latini, G., Di Nicola, G., & Pierantozzi, M. (2014). A critical survey of thermal conductivity literature data for organic compounds at atmospheric pressure and an equation for aromatic compounds. *Energy Procedia*, *45*, 616–625. https://doi.org/10.1016/j.egypro.2014.01.066

Steimer, S. S., Krieger, U. K., Te, Y. F., Lienhard, D. M., Huisman, A. J., Luo, B. P., Ammann, M., & Peter, T. (2015). Electrodynamic balance measurements of thermodynamic, kinetic, and optical aerosol properties inaccessible to bulk methods. *Atmos. Meas. Tech.*, *8*(6), 2397–2408. https://doi.org/10.5194/amt-8-2397-2015

Willoughby, R. E., Cotterell, M. I., Lin, H., Orr-Ewing, A. J., & Reid, J. P. (2017). Measurements of the Imaginary Component of the Refractive Index of Weakly Absorbing Single Aerosol Particles. *Journal of Physical Chemistry A*, *121*(30), 5700–5710. https://doi.org/10.1021/acs.jpca.7b05418

[revised manuscript text omitted]

It is important to note that as expected, for a pure PEG400 particle (i.e. no measurable absorption at 473 nm) at similar conditions, but a slightly less sensitive setup, no signal could be detected.

To further demonstrate the potential of the EDB-PPS approach in determining the imaginary RI with high sensitivity and precision, an additional particle from the same PEG400-CA batch, with radius of 12.858 µm was levitated and the response of the EDB to change in the net vertical force was recorded over about 16 hours of illumination cycles. To take advantage of the inverse pressure dependence of the photophoretic effect, this experiment was conducted at 400 hPa, which is at the lower limit of our experimental set up. Figure 6 shows the measured $\Delta U/U_0$ and the point-by-point retrieved $\kappa$, which decreases from about 1.25 × 10$^{-4}$ to about 0.06 × 10$^{-4}$ over the "laser on" time (i.e. 25% of the total experiment time). Both traces demonstrate a decay feature as a result of slow photolysis of the CA (Jørgensen and Skibsted, 1991). The initial retrieved value $\kappa = (1.251^{+0.315}_{-0.213}) \times 10^{-4}$ is about 20% lower than the value retrieved from the bulk cuvette experiment [$\kappa = (1.394 \pm 0.05) \times 10^{-4}$] but with overlapping uncertainty range. The overall uncertainty in retrieved $\kappa$ demonstrated in Figure 6b is estimated to be up to 25% for $\kappa \approx 10^{-4}$ and up to 60% for $\kappa \approx 10^{-5}$. The gray shaded area on the $\Delta U/U_0$ plot (Figure 6a) is the uncertainty in each $\Delta U/U_0$ data point determined from the standard deviation on the value of $U$ averaged during "laser off" and "laser on" stages of each individual illumination cycle. These experimental uncertainties, with values of 2×10$^{-5}$ - 10×10$^{-5}$ (unitless) together with point-by-point variability of up to ±10$^{-4}$, are a result of the sum of system instabilities. These are independent of the value of $\Delta U/U_0$ and are a significant component limiting the sensitivity to $\kappa$ of this experimental approach. The experimental uncertainty contributes up to 10% uncertainty on the value of the retrieved $\kappa$ for values down to $\kappa \approx 2 \times 10^{-5}$ and about 30% for $\kappa < 2 \times 10^{-5}$. The uncertainty on the radiant flux used in the model calculation is the major limiting factor for the sensitivity of the experimental approach contributing about 15% to the uncertainty of the retrieved $\kappa$ down to $\kappa \approx 2 \times 10^{-5}$.

We determine the overall sensitivity of this approach to the imaginary RI (within the limitation of our experimental setup) to be in the range of 1 × 10$^{-5}$ - 2 × 10$^{-5}$. The accumulated uncertainty of $\kappa$ from all other parameters in Equations (14) and (17) is below 10% for $\kappa$ down to $\kappa \approx 2 \times 10^{-5}$. These include uncertainties in the particle size, real part of the CRI, density and thermal properties. However, the major contributor to this uncertainty is the range of selected values of the thermal accommodation coefficient mentioned in section 2.3. An uncertainty of 18% on the thermal accommodation coefficient ($\alpha_T = 0.85 \pm 0.15$) propagates to 4% − 7% uncertainty on the retrieved κ. This small contribution to the overall uncertainty demonstrates the usefulness of the EDB-PPS approach for the retrieval of the imaginary RI of organic particles with unknown

composition following accurate Mie resonance spectroscopy (for determination of size and real RI) as long as assumptions of
320 sphericity and homogeneity hold.

Additional potential source for uncertainty in case the particle is composed mostly of unknown or not well-characterized substances is the particle's thermal conductivity. A solution would be to consider that many organic compounds are very similar with respect to their thermal properties and use an approximated value with an appropriate uncertainty. Latini et al. (2014) published a literature survey of 4740 thermal conductivity data sets of organic compounds at atmospheric pressure and
325 reduced temperature of about $0.6 \pm 0.14$. A partial list of the data from this publication (excluding 1340 data sets of refrigerant compounds) is presented in Table 1. The list clearly show the similarity in thermal conductivity for many organic compounds abundant in atmospheric aerosols.

**4 Conclusion**

This study demonstrates the usefulness of the electrodynamic balance – photophoretic spectroscopy (EDB-PPS) technique to
330 retrieve the imaginary component of the complex refractive index of a slightly absorbing organic particle levitated in an environmental chamber. We showed agreement between measurements and model calculation and reliable retrieval of the imaginary RI at a wavelength of 473 nm, in the range of $10^{-5} \leq \kappa \leq 10^{-4}$ with uncertainty of about 25% for $\kappa = 10^{-4}$ and 60% for $\kappa = 10^{-5}$. The major limiting factor for sensitivity and precision within our setup is the uncertainty in the radiant flux.
335 In this study, we chose to emphasize the usefulness of EDB-PPS at the lower range of κ due to availability of other, simpler to implement, particle phase techniques suitable for retrieval of higher values of κ. Nevertheless, EDB-PPS is not limited to $10^{-5} \leq \kappa \leq 10^{-4}$. Figure 7 shows a simulated signal ($\Delta U/U_0$) over 6 orders of magnitude of κ, from $10^{-8}$ to $10^{-2}$, for a PEG400 based particle with a radius of 10 μm, At the lower end of this range (left side of figure 7a) κ is effectivity zero and the simulated signal is negative (i.e. the particle moves away from the light source) due to direct photophoresis (radiation pressure)
340 alone. Note how, as κ increases, the signal increases to positive values (figure 7a and 7b) due to an increase in the thermal asymmetry parameter that results in negative indirect photophoresis. With additional increase in κ, more energy is absorbed on the illuminated side of the particle. This offsets the hotspot of absorbed energy on the surface of the 'dark' side of the particle. As a result, the thermal asymmetry parameter and the magnitude of the signal are reduced. As κ continues to increase the asymmetry parameter changes sign from positive to negative as the illuminated side of the particle becomes warmer than
345 the 'dark' side. This is shown in figure 7c as negative signal and illustrated in 7d as the particle shifts from the regime described in the upper part of figure 7d to the one described by lower part. Based on this simulation and on our system's sensitivity limitations we determined the lower limit for retrieval of κ at $10^{-5}$ and do not determine an upper limit. We do however, note areas with high uncertainty, namely, around $\kappa \approx 10^{-3}$, where the change in signal flattens and around the point where the regime changes from negative to positive indirect photophoresis. The latter depends heavily on the particle size. An important caveat
350 is the non-injective behavior observed in figure 7b were two values of κ could solve for the same signal. To address the issue

one would need to have prior knowledge on the order of magnitude of the particle's absorption (for example; below or above $\kappa = 10^{-3}$). Alternatively, in aging experiments, as $\kappa$ evolves, the direction of the change of the signal would clarify the direction of the change of $\kappa$.

[revised manuscript text omitted]

590 **Table 1: thermal conductivity (W m⁻¹ K⁻¹) of organic compounds from Latini et al (2014) and references therein.**

|  | Number of data sets | Mean value | Standard deviation | Standard deviation % |
|---|---|---|---|---|
| Alcohols | 775 | 0.1482 | 0.0256 | 17.3 |
| Alkanes | 1025 | 0.1259 | 0.0182 | 14.5 |
| Alkenes | 135 | 0.1301 | 0.0244 | 18.7 |
| Aromatics | 570 | 0.1174 | 0.0180 | 15.3 |
| Carboxylic acids | 318 | 0.1576 | 0.0427 | 27.1 |
| Cycloalkanes | 35 | 0.1262 | 0.0134 | 10.6 |
| Cycloalkenes | 10 | 0.1303 | 0.0104 | 8.0 |
| Esters | 236 | 0.1261 | 0.0179 | 14.2 |
| Ethers | 111 | 0.1268 | 0.0181 | 14.3 |
| Ketones | 185 | 0.1417 | 0.0203 | 14.3 |

595

600

605

610

615

[Figure]

**Figure 1: Schematic diagram of the experimental set up. (a) Measurement of high-resolution elastic light scattering at 900 by illuminating the particle with a 765–781 nm TDL. (b) Continuous illumination of the particle with a red light-emitting diode (LED)**
620    **used for the active feedback adjustment of the DC voltage and alternating illumination with the 473 nm laser used to impose vertical optical forces on the particle.**

[Figure]

625 ____________________________________

Figure 2: change in the applied DC voltage in response to alternately illuminating a trapped light absorbing particle. Here, a particle
630 with 12.72 µm radius and $\kappa \approx 1.36 \times 10^{-4}$ was levitated in 400 mbar of dry nitrogen. For more details on how the particle size and
imaginary part of the CRI where determined refer to section 2.2 and 3.

[Figure]

635    **Figure 3. Indirect photophoretic force calculated with the Rohatschek (1995) approximation at the full pressure range (gray line), Mackowski (1989) analytical solution (dashed line) and the Soong (2004) correction (solid line) for a slightly absorbing particle with radius of 10 μm, T = 20 ⁰C, m = 1.466 + i10⁻⁴, λ = 473 nm, I = 35 mW mm⁻² and full thermal accommodation. The black rectangle and the insert plot show the pressure range that is possible with our experimental system.**

Figure 3. Indirect photophoretic force calculated with the Rohatschek (1995) approximation at the full pressure range (gray line), Mackowski (1989) analytical solution (dashed line) and the Soong (2004) correction (solid line) for a slightly absorbing particle with radius of 10 $\mu$m, T = 20 $^0$C, m = $1.466 + i10^{-4}$, $\lambda$ = 473 nm, $I$ = 35 mW mm$^{-2}$ and full thermal accommodation. The black rectangle and the insert plot show the pressure range that is possible with our experimental system.

[Figure]

Figure 4. Measured Mie scattering spectra at π/2 angle of a levitated particle. The positions of the TE and TM resonance modes (e.g. $TM_n^l$; identified by their order ($l$) and mode ($n$) numbers) were used to retrieve the particle's real refractive index ($n_D$ =1.4665, $m_1$ =2745) and size ($a$ = 9.2906 µm). We note that the lower order modes in the fitted TE spectrum are too narrow to be observed in the measured spectrum with the current resolution. Residual noise in the measured spectrum originate mostly from laser power oscillations due to frequency scanning operation and from horizontal oscillation of the particle due to the applied AC field.

645

[Figure]

650 **Figure 5. Measured and simulated EDB response ($\Delta U/U_0$) to the illumination of a slightly absorbing particle due to the photophoretic effect. The two analytical solutions; Mackowski (1989) dotted line and Soong et al. (2005) solid line that account for slip-flow conditions agree well with our measurements (empty circles). Error bars: standard deviation over five illumination cycles. Gray shaded area: measurement uncertainty propagated through the Soong model calculation, dominated by the 15% uncertainty on the radiant flux measured with a power meter and a beam profiler.**

655

[Figure]

[Figure]

660     **Figure 6. (a) Decay of the EDB photophoretic response (ΔU/U$_0$) during illumination of a slightly absorbing particle (PEG400 with 0.23% wt of carminic acid; CA). The decay is due to the slow photolysis of the CA. (b) The retrieved imaginary part of the complex refractive index. Gray shaded area in both plots represents the uncertainty in the measured and retrieved values.**

[Figure]

665

**Figure 7: simulated signal (and uncertainty) for a PEG400 particle, 10 μm in radius at T = 20 $^0$C, P = 400 mbar, $m_1 = 1.466$, $\lambda = $ 473 nm, $I = 35$ mW mm$^{-2}$ and full thermal accommodation with increasing imaginary part of the CRI. (a) Transition from negligible absorptivity and negative signal (i.e. dominated by radiation pressure) to positive signal (i.e. dominated by negative indirect photophoresis). (b) Increasing absorptivity leads to increasing positive asymmetry parameter and stronger positive signal**

670 **(i.e. negative indirect photophoresis). Additional absorptivity leads to asymmetry parameter of zero as the surfaces of the upper and lower hemispheres of the particle are effectively at equal temperature. (c) Additional increase in absorptivity leads to an increase of a negative asymmetry parameter and stronger positive indirect photophoresis. (d) An illustration of the positive and negative indirect photophoresis regimes. Black arrows represent the various forces acting on a 'strongly' (lower) and 'slightly' (upper) absorbing particle when illuminated with the laser beam (blue arrow).**